# Cell Senescence, Multiple Organelle Dysfunction and Atherosclerosis

**DOI:** 10.3390/cells9102146

**Published:** 2020-09-23

**Authors:** Gisela Machado-Oliveira, Cristiano Ramos, André R. A. Marques, Otília V. Vieira

**Affiliations:** CEDOC, NOVA Medical School, Faculdade de Ciências Médicas, Universidade NOVA de Lisboa, 1169-056 Lisboa, Portugal; cristiano.ramos@nms.unl.pt (C.R.); andre.marques@nms.unl.pt (A.R.A.M.)

**Keywords:** atherosclerosis, senescence, senescent organelles, cardiovascular diseases

## Abstract

Atherosclerosis is an age-related disorder associated with long-term exposure to cardiovascular risk factors. The asymptomatic progression of atherosclerotic plaques leads to major cardiovascular diseases (CVD), including acute myocardial infarctions or cerebral ischemic strokes in some cases. Senescence, a biological process associated with progressive structural and functional deterioration of cells, tissues and organs, is intricately linked to age-related diseases. Cell senescence involves coordinated modifications in cellular compartments and has been demonstrated to contribute to different stages of atheroma development. Senescence-based therapeutic strategies are currently being pursued to treat and prevent CVD in humans in the near-future. In addition, distinct experimental settings allowed researchers to unravel potential approaches to regulate anti-apoptotic pathways, facilitate excessive senescent cell clearance and eventually reverse atherogenesis to improve cardiovascular function. However, a deeper knowledge is required to fully understand cellular senescence, to clarify senescence and atherogenesis intertwining, allowing researchers to establish more effective treatments and to reduce the cardiovascular disorders’ burden. Here, we present an objective review of the key senescence-related alterations of the major intracellular organelles and analyze the role of relevant cell types for senescence and atherogenesis. In this context, we provide an updated analysis of therapeutic approaches, including clinically relevant experiments using senolytic drugs to counteract atherosclerosis.

## 1. Introduction

Atherosclerosis is a chronic inflammatory, progressive and age-linked condition that is associated with the development of major cardiovascular diseases (CVD), including ischemic heart and peripheral vascular disease as well as particular ischemic strokes [1]. Atherogenesis, the formation of atheromatous plaques in the arterial wall, is initiated by retention and extensive oxidative modifications of low-density lipoproteins (LDL) in the arterial *intima*, leading to endothelium activation and monocyte recruitment and migration across the endothelial barrier. Monocytes subsequently differentiate into a macrophage phenotype, accounting for phagocytosis and degradation of the excessive accumulation of oxidized LDL. The cellular metabolization pathways of macrophages eventually become overloaded, which inevitably lead to irreversible accumulation of lipids within the lysosomal compartment and disrupted lysosome function. As a consequence, macrophages transform into foam cells that are unable to contribute to additional lipid metabolization and initiate the secretion of pro-inflammatory factors that further aggravate the development of the atheromatous plaques. The accumulation of oxidized LDL to levels that overcome the metabolization capacity of macrophage and foam cell’s lysosomes thus appears a very relevant cellular event to trigger atherogenesis [2,3,4].

Along with the cellular differentiation of macrophages occurring in the arterial *intima*, vascular smooth muscle cells (VSMC) from the tunica *media* also migrate into the sub-endothelial region. These cells become highly proliferative and tend to surround the evolving inflammatory process, forming a fibrous cap that stabilizes the plaque. Nevertheless, fibrous cap formation by VSMC also involves the secretion of extracellular matrix proteins, which facilitate plaque rupture [5,6]. Moreover, in more advanced atherosclerotic plaques, VSMCs can develop a foam cell-like phenotype, aggravating the lesion [7,8,9]. Notably, the structure of the atherosclerotic plaques is defective in the long term and continue to progress to mature atheromatous plaques containing a large necrotic core and a thin fibrous cap that over time becomes increasingly susceptible to rupture [10]. Depending on the size of the plaque and the capacity to enclose the inflammatory process, the rupture of the fibrous cap can lead to vessel thrombosis and the potentially deadly acute vascular diseases [10].

As an age-related disease, atherosclerosis is associated with a number of biological processes including cellular senescence [11]. Overtly, the multiple senescent cell types present in the vasculature were reported to ease various pathophysiological processes in atherosclerosis [12], with the senescence-associated pro-inflammatory phenotype (SASP) gradually contributing to atherosclerotic plaque progression and destabilization [13,14]. In atherosclerotic lesions, cellular senescence is driven not only by the exhaustion of replicative potential that is normally associated with aging, but also by a variety of other cellular stressors such as oxidized LDL [stress-induced premature senescence (SIPS)] [15], processes that will be further detailed in the next section. The inter-related sequence of senescence-driving events governing atheroma progression are represented schematically for clarity (Figure 1). In this review, we present the existing evidence regarding the main senescence cellular alterations taking place during atheromatous plaque maturation, with particular focus on the role of specific organelles in the senescent processes associated with atherosclerosis development. Lastly, we discuss potential strategies to circumvent cellular senescence and highlight the improvements still required in the field in order to develop novel therapeutic approaches for cardiovascular disorders.

## 2. Cell Senescence Overview

Cellular senescence is classically characterized by an irreversible cell cycle arrest that might be prompted by DNA damage, oxidative stress, nutrient deprivation, oncogenic insults or chemotherapeutic-induced toxicity. In addition, the irreversible cell cycle arrest in senescent cells is primarily imposed by an upregulation of the cell cycle inhibitors p16, p21 and p53 [16,17]. Senescence is associated with cellular alterations such as cytoplasm enlargement as well as irregular and flat morphology, distinctive features from those of proliferating cells. Likewise, functional modifications including increased resistance to apoptosis and development of the typical SASP, also take place during the process of replicative senescence, whilst maintaining metabolic activity. Particularly relevant, SASP is associated with the release of a number of factors including bioactive lipids, extracellular vesicles, proteins and nucleotides that mediate core pathophysiological events in senescence. More specifically, SASP is involved in the recruitment of immune cells, modification of the extracellular matrix and supports the maintenance of the cell-cycle arrest [17]. Indeed, the cellular secretions of the SASP act as paracrine and autocrine mediators that promote inflammation, aberrant communication between surrounding cells and ultimately lead to defective tissue remodeling [18,19,20]. SASP elements were reported to stimulate the immune system for senescent cell clearance, a protective process that eventually becomes inefficient with aging, resulting in intensified inflammatory responses and chronic inflammatory disorders [21,22]. As such, senescent cells tend to chronically accumulate in multiple aged tissues [18], indicating that senescence may contribute to regulating the non-pathological age-associated events. In addition, different studies have pointed to cellular senescence as being involved in a number of age-related diseases, such as atherosclerosis, cancer, hepatic steatosis, Alzheimer’s disease, fibrotic pulmonary disease and osteoarthritis [23,24,25,26,27,28], which indicates that cell senescence may be entailed in driving these diseases. Specifically, senescent cells have been found to accumulate in sites of atherosclerosis ex vivo in human samples [29,30], again suggesting the involvement of cell senescence in sculpting this disease. Studies conducted in LDL receptor deficient-mice have also shown that the removal of p16^INK4a^-positive senescent cells from atherosclerotic plaques contributed to repressing the typical pathological modifications [13], and can account for the suppression of the aging phenotype in different organs, including in the murine heart [23]. Nevertheless, cell senescence is also characterized by beneficial physiological functions, including its participation in cancer protection, as it is a way to avoid transmission of genetic and organelle damage through cell proliferation, and its role in the guidance of early embryonic development and in tissue repair [31].

In contrast to SASP, SIPS manifests upon exposure to exogenous cellular insult, such as oxidative stress. Both replicative senescence and SIPS prompt the same cell phenotype, and are regulated by some common signaling pathways, though the telomere length appears not to be altered in SIPS, thus conferring potential reversibility on this process [15]. Specifically, replicative senescence and SIPS can be mediated via the p53/p21 or p16 signaling pathways, depending on the type of cell and the species involved [32,33]. Telomere shortening activates either the p53/p21 or p16 pathways in humans, while only the p53/p21 pathway is induced in mice [34]. Still, it is generally established that telomere shortening and DNA damage lead to the induction of the p53/p21 pathway, while oxidative stress and general cellular stressors activate the p16 signaling pathway [32,33,35]. On the other hand, oncogene-induced senescence (OIS) implies cell proliferation arrest due to the activation of oncogenes such as Ras or pRB [27,36]. Importantly, additional studies have shown that senescence is not restricted to proliferating cells. Post-mitotic cardiomyocyte senescence was recently reported to occur independently of telomere shortening and damage [37]. In particular, senescent cardiomyocytes were shown to contribute to SASP, promoting cardiomyocyte hypertrophy in mice [37]. Senescent human cardiac progenitor cells were also reported to accumulate in failing hearts with advanced age, compromising heart regeneration due to SASP [38]. Importantly, amongst the different types of cardiac senescent cells, cardiomyocytes, as the main functional units of the heart, greatly contribute to cardiac senescence, compromising myocardial performance. Senescent cardiomyocytes were shown to exhibit functional decline, including reduced contractility, mitochondrial dysfunction and increased size. As a result of senescent cardiomyocytes accumulation with age, chronic inflammation is promoted, leading to cell death [39,40]. Interestingly, Meis1, a senescence-associated gene, was shown to contribute to the regulation of hematopoietic stem cell metabolism and oxidant stress response, via the transcriptional regulation of hypoxia-inducible factors [41]. More recent studies revealed that silencing senescence-mediated pathways through inhibition of Meis2 and Rb1 in adult rat cardiomyocytes leads to cardiomyocyte proliferation and cardiac repair upon myocardial infarction [42]. In addition, FoxO3 was recently reported to mediate cardiac protection from oxidative stress, counteracting cardiac dysfunction, apoptosis and senescence in mice [43]. Additional studies have revealed that melatonin inhibited cardiac progenitor cell senescence in response to oxidative stress, via H19/miR-675 pathway [44]. Moreover, the human amniotic fluid stem cell secretome was shown to antagonize senescence of mouse cardiomyocytes and human cardiac progenitor cells, via the modulation of NF-kB-mediated transcription [45]. Other studies also focus on the description of mechanisms fueling senescence in post-mitotic cells and its contribution to aging and tissue degeneration [46,47], but further research is needed to fully clarify the post-mitotic cell senescence process.

In addition, it has been reported that all senescent cells share a common secretory profile and that each particular senescent cell type also tends to express more specific secretory factors (see Table 1 to compare differential secretome profiles between cell types) [48]. Accordingly, senescent human fibroblasts and epithelial cells were shown to express different SASPs in culture. Human epithelial tumor cells also acquire the SASP and express specific secretions (in vivo) after being exposed to DNA-damaging chemotherapy. Moreover, the malignant epithelial–mesenchyme transition (EMT), which confers invasive and metastatic properties on epithelial cells, was demonstrated to be stimulated by senescent cells, via the SASP interleukins IL-6 and IL-8. Remarkably, both the oncogenic RAS and the loss-of-function of the tumor suppressor p53 caused amplification of the SASP, which suggests that senescent cells may provide a tissue microenvironment that supports different stages of tumor progression [48]. Taken together, the conserved secretory profile and the more intrinsic secretory factors related to the particular cell genotype and tissue of origin suggest that the SASP is not a steady phenotype and continues to intervene actively in numerous pathophysiological processes.

## 3. Endothelial, Smooth Muscle and Immune Cells: Major Senescence Traits Implicated in Vascular Disease

Studies including post-mortem histological analysis have provided clinical data demonstrating considerable accumulation of different types of senescent cells in atherosclerotic arteries in comparison to aged healthy vessels [14]. Endothelial cells (ECs), which form the endothelium, are located on the arterial *intima* and maintain vascular homeostasis, being entailed in several biological functions, such as blood pressure regulation, coagulation, angiogenesis and inflammatory response [49]. Importantly, ECs are continuously subject to hemodynamic forces, being capable of remodeling their cytoskeletal structure and evoke adaptive signaling pathways to minimize intracellular stress and maintain vascular homeostasis. Nonetheless, pathological mechanical forces or excessive inflammation leads to perturbed EC responses, decreasing nitric oxide synthesis, increasing adhesion molecules expression and oxidative stress, resulting in EC dysfunction [50], which frequently leads to CVD. However, depending on a number of predisposing factors such as advanced age, elevated salt intake, hypercholesterolemia, hyperglycaemia and hypertension, the mechanisms underlying the endothelial dysfunction can be markedly different [49]. In particular, age is a critical cardiovascular risk factor contributing to the development of atherosclerosis [51], with senescent ECs being frequently found in atherosclerotic plaques of elderly patients [52,53,54]. In this instance, EC senescence has been proposed as a pathophysiological contributor to developing vascular disease [27]. Further confirming this concept, active proatherogenic mechanisms are present in senescent cells of aging vessels and senescent ECs have been shown to present lower NO generation and increased expression of the adhesion molecules VCAM1 and ICAM1, which bind monocytes prompting endothelium infiltration [55]. In vitro studies have also revealed that senescent ECs are more prone to undergoing apoptosis and show compromised formation of tight junctions, which can enhance the retention of oxidized LDL in the arterial *intima* and contribute to atherogenesis [56,57]. In agreement with this, monocytes have been reported to have higher binding affinity to senescent ECs, shown in X-ray-induced or replication-induced senescent ECs [58], suggesting that ionizing radiation can cause CVD. Moreover, at the vessels’ bifurcation sites, where blood flow is disturbed, ECs exhibit shorter telomeres, indicating senescence exacerbation. In addition, senescent ECs induced by disturbed flow display reduced migration and altered expression of inflammatory factors, which may be due to the activation of p53/p21 signaling [59]. Consistently, protection against oxidative stress and age-associated endothelial senescence was reported to be achieved via the inhibition of NF-kB signaling [60], a potent promoter of p53-mediated SASP in human and rodent cells [61,62,63]. Furthermore, ECs produce extracellular vesicles, the endothelial microvesicles, which act as extracellular mediators that contribute to maintaining the vessel homeostasis [64], and whose functional disruption has been related with endothelial senescence. Senescent cells can develop the SASP and increase the secretion of both soluble factors and extracellular vesicles that act as conveyors of senescence signals [65]. Indeed, various studies have shown that extracellular vesicle secretion increases during senescence [66,67,68]. These facts point to the microvesicles as being part of the SASP, potentially constituting a mechanism to allow the release of insoluble proteins and the activation of specific signaling pathways in the target cells [69], possibly playing a role in the atherosclerotic plaque developmental process. In addition, microvesicles from elderly individuals’ plasma or senescent ECs were reported to stimulate the calcification of human aortic SMCs. Not only the microvesicles in the plasma rise with aging, but also the senescent EC-secreted microvesicles contain higher quantities of calcium and calcium-binding proteins, known to participate in vascular calcification, indicating that microvesicles could be used as markers of vascular calcification in atherosclerotic plaques [67].

The arterial *media* is formed in part by VSMCs that synthesize the extracellular matrix and are responsible for the vessel contraction [70]. A number of studies have provided evidence for the occurrence of VSMC senescence in atherosclerosis. VSMCs derived from human atherosclerotic plaques exhibit a lower level of proliferation [71], higher p16 and p21 expression levels and a larger and flattened morphology in comparison to cells from healthy arterial *media* [72]. Replicative VSMC senescence, reported to occur in human atherosclerosis, was associated with decreased telomere length in the more advanced and severe atherosclerotic processes, possibly due to higher rates of VSMC proliferation during early lesion development. Still, oxidative stress also causes telomere shortening and VSMC senescence, contributing to the acceleration of the atherogenic process [29]. In particular, pro-oxidants have been postulated to trigger SIPS in human VSMCs [73], whereas chronic oxidative stress accelerates replicative senescence [29]. Additionally, there are reports of DNA damage-induced VSMC senescence in atherosclerotic mice [74]. Not only a reduction in telomere length, but also the loss of telomere binding factors and modifications in the telomere structure may trigger VSMC senescence in atherosclerosis [75]. Apart from the changes in senescent VSMC proliferative potential, which can prompt atherosclerotic plaque instability due to the reduction in VSMCs in the fibrous cap, other studies indicated that VSMC senescence promotes plaque destabilization via stimulation of inflammation. Senescent human VSMCs release SASP mediators and display reduced expression of anti-inflammatory factors. SASP mediators’ secretion induces chemotaxis of monocytes and stimulates neighbor non-senescent VSMCs and ECs to secrete cytokines and express adhesion molecules, enhancing disease progression. Moreover, senescent human VSMCs produce lower amounts of collagen than normal VSMCs, which can increase plaque vulnerability [76]. In addition, VSMCs may additionally assume an “osteoblast-like” phenotype, increasing their susceptibility to calcification, during replicative senescence events [77,78], which is related with cardiovascular complications and possibly also contributing to plaque vulnerability [79]. Even though most available data focus on the role of senescent VSMCs in advanced atherosclerotic lesions, there are also reports demonstrating their potential involvement during early plaque development [80], emphasizing the importance of normal VSMC function in protecting the arteries against atherosclerosis. Autophagy, a cellular mechanism that allows removal of unnecessary or dysfunctional components, was shown to protect VSMCs from cell death and to suppress VSMC senescence, upon moderate activation [81,82,83]. Therefore, the imbalance amongst these pathways possibly fuels the development of unstable atherosclerotic plaques [84].

Immune system cells, particularly the senescent immune cells found in the vasculature wall, also contribute to the development of atheroma. Indeed, different studies have revealed that senescent leukocytes contribute to the progression of atherosclerotic plaques and senescent effector memory T (T-EMRA) cells were identified in unstable atherosclerotic plaques [85]. For instance, aged individuals presenting leukocytes with shortened telomeres were reported to have higher mortality, at least in part due to heart disease [86]. Most interestingly and potentially clinically relevant, leukocyte population analysis indicates that telomere shortening allows for the prediction of atherosclerosis and CVD [87,88], and CD4+ as well as CD8+ T-EMRA cells are considered predictors of cardiovascular-related mortality in older individuals [89]. In addition, patients with advanced atherosclerosis have been shown to display monocytes that tend to produce elevated levels of ROS and pro-inflammatory cytokines [13,90]. Moreover, these pro-inflammatory phenotypic alterations in macrophages were shown to be driven by cellular senescence [91]. Transgenic and pharmacological approaches also revealed that subendothelial senescent foam macrophages present in early lesions enhance tumor necrosis factor (TNF)-α-mediated Vcam1 expression and the monocyte chemoattractant protein-1 (MCP-1) gradient, to maintain monocyte recruitment to the site of lesion [13].

## 4. Cellular Organelles in Senescent Cells

Senescent cells exhibit morphological and functional defects, which derive from intracellular alterations in their morphology, function, level of development and mass of their organelles. Independently of the specific intracellular organelle, senescence imposes common alterations including functional deficiencies, an increase in total organelle mass and abnormal intracellular signaling [92]. Classic biological senescence markers point to organelle defects, as detailed below. Briefly, dysfunctional lysosomes commonly exhibit increased senescence-associated β-galactosidase (SA-β-gal) activity, whereas defective oxidative phosphorylation in mitochondria results in increased reactive oxygen species (ROS) production. In addition, senescence-related alterations in the endoplasmic reticulum (ER) lead to the unfolded protein response (UPR), along with the DNA damage and the senescence-associated heterochromatin foci (SAHF) in the nucleus. Although it is still not completely elucidated how these intracellular alterations in morphology and function take place and which regulatory pathways induce multiple organelle dysfunction, they are likely related, either by consequence or predisposal, to the defects in cellular metabolism detected during senescence [93]. One leading hypothesis is that the increased organelle production by senescent cells may occur as a compensatory response for the defective organelle function caused by the oxidative stress. Still, the lately produced organelles potentially worsen senescence due to the continuous exposure to ROS [93].

In normally functioning cells, the proteome’s integrity is maintained via a concerted regulation between autophagy, proteasome-dependent degradation and chaperone-mediated protein folding. These regulatory mechanisms also appear to be impaired during cellular senescence as protein damage and aggregates and abnormal protein synthesis are highly prominent [94]. In addition, macroautophagy, which controls the elimination of damaged organelles resulting from oxidative stress, is frequently not operating adequately, leading to further deregulation of organelle homeostasis [95,96]. Therefore, the increased organelle mass that characterizes senescent cells is likely to be related to mitochondrial dysfunction, anomalous anabolic activities and defective autophagy, which in turn result in additional oxidative stress and aggravated deregulation of organelle biogenesis.

### 4.1. Mitochondria and the Lysosomal Compartment

Amongst all cellular organelles, mitochondria and lysosomes show the most pronounced changes during cell senescence [97,98,99]. Mitochondria are the main source of cellular ATP [100], displaying fusion cycles to mix the contents of partially damaged organelles [101,102], and fission cycles to generate new mitochondria and enable the degradation of dysfunctional organelles by autophagy [103]. Mitochondrial DNA damage has been stated to be potentially involved in cellular senescence [104], and a number of studies revealed mitochondrial oxidative phosphorylation dysfunction’s involvement in different senescence cellular models [105,106,107,108,109,110,111]. Dysfunctional mitochondria are thus a hallmark of cell senescence, which was reinforced by another study involving targeted depletion of mitochondria, also demonstrating that mitochondria contribute to regulate the SASP and to promote the cell-cycle arrest in senescent cells. In this context, ROS generation by mitochondria and the consequent DNA damage response were shown to contribute to develop the senescent phenotype, while the mechanism underlying the cell-cycle arrest is still not completely understood [112].

Phenotypic and morphological alterations in senescent mitochondria include organelle enlargement [113,114,115] and increased mitochondrial cell mass [115,116]. Although the mechanism and the regulatory pathways of the mitochondrial senescence are not completely clarified, it has been proposed that damaged enlarged mitochondria may accumulate throughout senescence because of complications during autophagic degradation [117,118]. In particular, mitochondrial fission activity is downregulated during cellular senescence [119], potentially decreasing mitochondrial autophagic turnover (mitophagy) [115], thus incrementing mitochondrial mass. Likewise, an inefficient synchronization between autophagy, mitochondria fission and fusion also appears to take place in senescent and aged cells, leading to the accumulation of damaged mitochondria [120]. In agreement with this, blocking mitochondrial fission triggers a senescence-like state, with the formation of elongated mitochondria and enhancing the generation of ROS [115,121]. Adding to the decrease in autophagic turnover, defective autophagic processes also result in compromised formation of autolysosomes and reduced digestion, leading once more to the pathological accumulation of dysfunctional mitochondria. Taken together, it appears clear that senescent cells may become unable to control the mitochondrial mass, leading to a progressive increase in the mass of defective mitochondria per cell [93], which can consequently give rise to premature senescence or even apoptosis [122]. Importantly, impaired mitophagy has been related with the development of age-associated cardiovascular pathologies. Accordingly, the mitochondrial enzyme monoamine oxidase-A (MAO-A) has been reported to drive SIPS in cardiac cells via mitophagy impairment and consequent mitochondrial dysfunction and chronic ROS generation [123].

Similar to that proposed for the other organelles, it has been hypothesized that the increase in mitochondrial mass occurs as a cellular compensatory mechanism for the functional mitochondria decay [124,125]. This is particularly relevant for the case of mitochondria that possess their own mitochondrial DNA and retrograde signaling pathways to the nucleus. Indeed, cellular compensatory mechanisms for mitochondrial proliferation and potentiation of oxidative capacity are well known and frequently reflected in disorders associated with mitochondrial damage, like the CVD [126,127,128]. Amongst the mitochondrial retrograde signaling events, transcriptional reprogramming and modulation of different cellular functions are entailed in changing the cell fate to proliferation, senescence or death [129,130,131,132,133]. ROS, one of the mitochondrial retrograde signaling messengers, is known to damage DNA, lipids and proteins, leading to the activation of damage responses. Nevertheless, mitochondria and the mitochondrial DNA are particularly vulnerable to damage, as the electron transfer chain is the main site for ROS production. This, combined with the high number of mitochondria present in cells, makes these organelles remarkable in disseminating oxidative stress and prompting senescent phenotypes [134,135,136,137]. Oxidative stress causes DNA damage by oxidizing nucleotide bases and inducing single- (SSB) and double-strand breaks (DSB). DSB stimulate a DNA damage response and the expression of the cell-cycle inhibitors p53 and p21, which mediate senescence, and result in the staining of the SAHF. DSB-mediated DNA damage response, also called telomere-independent premature senescence, in contrast with the SSB, to which the telomeres are susceptible due to the enrichment in guanine triplets, indicating that the mitochondria-induced oxidative stress may contribute to define the telomere status [138,139].

Remarkably, senescent cells were reported to extrude fragments of chromatin from the nucleus into the cytoplasm [140]. This cytoplasmic chromatin fragments (CCFs) were stated to induce the SASP during cell senescence [141,142,143]. In this context, a recent study revealed that the formation of CCFs and thus the activation of the SASP are initially triggered by mitochondria-nuclear retrograde signaling pathways, possibly prompted by increased ROS production in dysfunctional mitochondria [144].

Mitochondrial defective function and related ROS generation can also cause lipid oxidation and lipid deposits [25,112], as well as lipofuscin accumulation [145]. Although the accumulation of lipids is known to occur in senescent cells, the characterization of such lipids, the alterations of lipid metabolism and their role in senescence and atherosclerosis are still very poorly elucidated. Despite the various methods that can be used to detect the lipid profile in cells, senescence-associated lipidomics is highly variable. A significant variation of lipid metabolites between replicative senescence and OIS is an example of the great differences in lipid metabolite composition in senescent cells. In addition, these differences in the metabolic profile appear to be partly explained by the initial trigger and the type of cell involved [146]. Considering the central lipid deposition events driving the formation of atheromas and the associated involvement of cell senescence, it is clear that further clarification of the lipid metabolism in senescent cells is imperative to better elucidate atherogenesis. Additionally, although a great deal of investigation is clearly required, the high variability in the composition of intracellular lipids between adult cells and cells enduring pathological processes opens the possibility to unravel the lipid profile of different organelles or a bulk lipid profile that can be used as a biomarker of senescence.

Lysosomes are involved in several cellular functions, being considered the main catabolic organelles [147], containing hydrolytic enzymes and highly acidic conditions to degrade a multiplicity of substrates [148]. A number of in vitro and in vivo studies revealed an increased lysosomal mass in senescent somatic cells and in aged post-mitotic cells [149,150,151,152,153]. The observed increase in the lysosomal mass was reported to be mostly due to an increase in the number of lysosomes containing an indigestible cargo such as lipofuscin, occasionally called residual or dense bodies [154]. Remarkably, lipofuscin was shown to induce the expression of Bcl-2, which confers resistance to apoptosis, leading to the hypothesis that the accumulation of lipofuscin can also be involved in the regulation of cellular senescence [155]. In contrast, lysosomes accumulate oxidized LDL-derived lipids in the macrophages and VSMCs at atheromatous sites [3,4,7]. In addition, several research lines indicate that lysosome biogenesis increases during cellular senescence [121,156,157,158,159,160], which is usually supported by elevated SA-β-gal activity, a marker of increased lysosomal number or activity [161]. The increase in the lysosomal mass observed during cellular senescence is thus associated with progressive structural alterations and accompanied by changes in lysosomal activity. Intrinsically related with this is the modification of the lysosomal pH that alters the activity of the majority of the lysosomal enzymes [162] and compromises the substrates of autophagic degradation as well as the endocytic cargo in senescent lysosomes [163,164,165]. Most interestingly, it has been recently shown that lysosomal oxidation of aggregated LDL alters the lysosomal pH, inducing cellular senescence and increasing secretion of pro-inflammatory cytokines in human macrophages [166]. These findings reinforce once more the active contribution of senescent lysosomes and the process of cellular senescence for atherogenesis. Other relevant functions of lysosomes during cell senescence include the clearance of the CCFs and the anabolic metabolization of chromatin. Indeed, it has been reported that CCFs extruded from the nucleus of senescent cells were targeted to be processed by the lysosomal/autophagy machinery [140]. According to the data gathered, other authors have demonstrated that senescent lysosomes exhibit reduced capacity to sense and respond to internal and external stimuli, which reflects their central role in the integration of multiple signals, governing cellular senescence and aging [167]. In this context, it is important to stress the involvement of the lysosomal/autophagy machinery in regulating the inflammasome, a protein complex that is involved in the secretion of some inflammatory cytokines. Specifically, defective autophagic clearance of inflammasomes has been shown to be implicated in the development of cardiometabolic disease [168], further reinforcing the consequences of the lysosomal/autophagy system being compromised.

Most interestingly, senescent cells have been shown to activate the mechanistic target of rapamycin complex 1 (mTORC1), a lysosomal serine-threonine kinase that regulates autophagy, protein synthesis, metabolism, cell growth and survival [169,170,171,172], these processes being specifically inhibited by rapamycin [173]. The mTORC1 activation in senescent cells may arise from defects in amino acid and growth factor sensing, partly due to the depolarization of senescent cell plasma membrane, compromising the inhibition of growth factor signaling [174]. mTORC1 induction in senescent cells may also contribute to developing the SASP, as mTORC1 inhibitors seem to reduce inflammation instigated by senescent cells and consequently the SASP [175]. Agreeing data were provided, indicating that these mTORC1 inhibitors operate via different mechanisms to inhibit the SASP, contributing to the up-regulation of DNA repair proteins and associated post-transcriptional modifications [174,176]. Importantly, inhibition of the mTOR pathway has already been reported to result not only in anti-senescence but also in anti-atherosclerotic effects [177,178,179].

Apart from the well-established crosstalk between mitochondria and the nucleus, several lines of evidence reveal the importance of a lysosomal–mitochondria axis in the regulation of cellular senescence. Coupling mitophagy and autophagy induction has been suggested to underlie the significance of the lysosomal-mitochondria axis in the control of cell senescence [97]. Accordingly, re-establishment of the lysosomal-mitochondria axis was reported to cause the reversion of senescent to juvenile phenotypes [98]. Indeed, studies in *C. elegans* have also shown that mitochondrial functional recovery is triggered by lysosomal acidification, reflecting the implication of the lysosomal-mitochondria axis in attenuating senescence [99]. As such, amongst the complex organelle interplay in cells, the lysosomal–mitochondria axis appears to be emphasized due to its role in regulating cell senescence.

### 4.2. Peroxisomes, Cytoskeleton and Nuclei

Peroxisomes are highly dynamic organelles, exhibiting variations in size and morphology, but also in cellular abundance and function, depending on external stimuli. Importantly, peroxisomes enclose numerous enzymes that regulate lipid metabolism and ROS signaling [180]. Peroxisomal ROS have homeostatic signaling roles, and the wide panel of peroxisomal antioxidant enzymes, including the predominant catalase, controls their effects. ROS produced by peroxisomes crosstalk with mitochondrial ROS signaling [181], being thought to be in excess in cardiometabolic diseases [182]. Aging, a crucial risk factor for cardiometabolic diseases, is strongly associated with an accumulation of dysfunctional peroxisomes [183]. Human patients affected by an inherited catalase deficiency were reported to have increased risk to develop age-related diseases, including atherosclerosis [184]. Accordingly, studies conducted in high-fat diet mice models revealed that catalases may improve atherosclerosis [185], and enhanced activity of catalases was observed in *foamy* cells from rabbit aorta atherosclerotic lesions [186]. Notably, peroxisome-mediated oxidative stress was suggested to contribute to the initial stages of peroxisome dysfunction and cellular senescence in human fibroblasts [187]. Agreeing later studies performed in a model of human cellular aging also revealed that the restoration of the oxidative balance in peroxisomes is associated with a reduction in cellular senescence [188]. As such, peroxisomal homeostasis maintenance via coordination between peroxisome biogenesis and peroxisome selective autophagic degradation (pexophagy) is vital to avoid pathologies such as atherosclerosis, which are related with the accumulation of dysfunctional peroxisomes [180].

The cell cytoskeleton plays an important role in various processes of non-senescent and senescent cells, including the maintenance of cell morphology, cell division, motility and intracellular trafficking. Not surprisingly, senescent cells exhibit alterations in these cellular processes that dictate cytoskeleton changes at the structural and functional levels. Different studies using various types of senescent cells have indicated that increased levels of the intermediate filament protein vimentin contribute to determining the senescent morphology [189,190]. Moreover, it has been reported that vimentin is prone to alterations by advanced glycation endproducts (AGEs), suggesting that high levels of vimentin or glycated vimentin may drive pathologies associated with aging. According to this, vimentin accumulation was reported to be related with cataract formation [191]. Regarding the actin filaments, vital to determining cell shape and motility, some changes have been reported to occur during replicative senescence [192,193,194] but lacking consistency, which may be due to the differential regulation of actin isoforms in the various tissues or be related with variations in the levels of expression with aging [195]. In addition, microtubules, key modulators of intracellular transport, cell division and polarity, were shown to greatly increase during cellular senescence, and multiple microtubule organizing centers (MTOC) were found in some cells [196]. On the contrary, others have presented evidence of decreasing levels of tubulin expression in senescent or aged cells [192,193]. Despite the described evidence linking cellular senescence and modifications in the cytoskeleton during the aging process, studies are required to understand senescence-associated alterations in the cytoskeleton in the context of atherosclerosis.

On the other hand, nuclear changes are well documented in senescent cells, which show prominent and at times multiple nuclei, with severe chromatin condensation detected as large fluorescent punctae in senescent nuclei after staining with 4′,6′-diamidino-2-phenyl-indole (DAPI). These structures, called SAHF, are not exhibited in the nuclei of proliferating cells, where a homogeneous fluorescent staining is observed [197]. In heterochromatin foci, DNA associates with proteins like the heterochromatin protein 1 (HP1) and the methylated lysine 9 histone H3 (H3K9me), being transcriptionally inactive [197,198]. Importantly, genes located in the SAHF, such as the E2F-responsive genes cyclin A and E, which allow cell cycle progression, are not expressed. Remarkably, ectopic overexpression of E2F1 in cells committed to senescence does not turn on the expression of its target genes, showing that SAHF dictates the permanent exit of the cell cycle during cellular senescence [197]. In addition, nuclear lamina assembly is modified in senescent cells, impacting on nuclear stability. In particular, down-regulation of lamin B1, a component of the nuclear lamina, has been reported as a hallmark of senescence in human and murine fibroblast cell lines, suggesting that it could be used as a senescence cell biomarker [199].

Instead, telomere shortening is accompanied by the loss of protective proteins, such as DS-, SS-binding and repair proteins, commonly associated with telomeres. A DNA damage response triggered at short telomeres is verified by the presence of DNA damage response factors, including _ϒ_-H2AX, at these chromosome regions [200], called telomere dysfunction-induced focus (TIF), being considered a telomere dysfunction marker. Notably, it was reported that _ϒ_-H2AX foci form at DNA DSB [201], and subsequent studies have shown the link between TIF and cellular senescence. Therefore, TIF has emerged as a potential marker for telomere shortening-induced senescence [202,203,204].

### 4.3. Golgi Complex and the Endoplasmic Reticulum

The Golgi complex is made up of a series of flattened and interconnected cisternae arranged around the MTOC in the perinuclear region of cells. The highly polarized Golgi apparatus exhibits the *cis* Golgi network (CGN), receiving cargo from the ER for processing, and the *trans* Golgi network (TGN), where the cargo exits for export. Both the cytoskeleton and the Golgi matrix proteins contribute to preserving the distinctive Golgi morphology [205,206]. Despite the significance of the Golgi complex in regulating various cellular functions, including intracellular transport and involvement in secretory pathways [207], knowledge on its structure in senescent cells was not clearly elucidated until the last decade. Golgi complex was then reported to present a compact structure either in non-senescent and pre-senescent cells, becoming dispersed in senescent cells, suggesting that the Golgi complex structure could be used as another cellular senescence marker [208]. Modifications of the Golgi apparatus morphology were also reported in senescent human cell lines and in aged mouse brain [209,210]. In agreement with this, recent findings demonstrate a disrupted Golgi apparatus morphology and alterations in the transcription of genes involved in Golgi architecture and function in senescent human dermal fibroblasts. Specifically, the Golgi apparatus presented a small and compact morphology in the perinuclear region of non-senescent cells, whereas a large and expanded structure throughout the cytoplasm was displayed by senescent cells [211], indicating a link between Golgi complex functions and the aging process. Importantly, hypertrophy of the Golgi complex and TGN has been reported to be correlated with lysosomal lipid accumulation from oxidized LDL in pigeon and human macrophages, similar to that seen in foam cells of atherosclerotic lesions [212]. These findings point to the need to further elucidate the link between modifications of the Golgi complex structure and function along cellular senescence, and its contribution to atherogenesis.

On the other hand, the ER is an organelle involved in protein folding and post-translational modifications, but also in the regulation of the calcium storage and secretory pathways [213]. The cellular response to ER stress is the UPR, to prevent the accumulation of un/misfolded proteins, relieving ER stress and reestablishing ER homeostasis [214,215]. In contrast, when cells cannot manage the stress and the ER becomes excessively overloaded, the UPR induces autophagy [216] or even apoptosis [217]. Both ER stress and UPR activation were reported to occur during cellular senescence [218], and to be particularly relevant in the replicative senescent phenotype [218,219,220]. The UPR activation and enlarged ER were also observed in melanocytes undergoing OIS [221]. In addition, the ER expansion and biogenesis were postulated to occur as a cellular attempt to adjust to the increased protein folding demand [222,223]. Later studies revealed that the ER expansion occurs concomitantly with cytoskeleton modifications and subsequent cell size increase [219,224], once again reflecting the interplay between different cellular components. Importantly, convincing findings have also shown that ER stress can prompt inflammatory responses in several age-associated diseases, including atherosclerosis [225]. In particular, the incidence of chronic endothelial ER stress and activation of the UPR have been demonstrated in vivo at arterial sites susceptible to atherosclerosis [226]. Figure 2 illustrates the main morphological and associated functional differences amongst senescent and proliferating cells, also contextualizing the numerous organelle modifications that occur along with the process of cellular senescence.

## 5. Assessment of Cellular Senescence

Despite the increasing number of studies in the context of cellular senescence, the lack of specific biological markers to enable the direct detection of senescence has limited the contribution of senescence identification for the clinical diagnostic of senescence-associated diseases. Even the current research of the biological processes involved in cellular senescence is supported by biological markers that allow only indirect identification of senescent cells. The most relevant methodologies currently available to assess cellular senescence are listed in Table 2.

To date, the biomedical research on senescence has had to exploit a combination of senescent cell biomarkers to avoid an imprecise or artefactual detection of senescent cells. Replicative senescence and SIPS confer an identical phenotype, except that SIPS is not associated with telomere attrition, thus being a reversible and not programmed biological process [15]. Detection of cell proliferation markers such as ki67 and the proliferating cell nuclear antigen (PCNA), as well as the incorporation of labelled nucleoside analogues into replicated DNA, are generally used to exclude proliferating cells from analysis. For this purpose, flow cytometry can be performed after immunofluorescence staining of nuclear antigens and BrdU incorporation [227,228]. Still, the lack of proliferation markers does not guarantee cellular senescence, as this is a common feature of quiescent cells, entailed in a reversible cell-cycle arrest elicited by environmental cues [229,230]. Morphology assessment using microscopy techniques can easily be performed to identify the typical senescent cell morphological features, but again the technique cannot confirm senescence per se [228]. In particular, the lysosomal beta-galactosidase activity, regardless of being typically detected at pH 4, can also be detected at higher pH 6 in senescent cells. As such, the SA-β-gal activity is capable of revealing the lysosomal beta-galactosidase activity and thus the increase in the lysosomal mass in senescent cells [161]. Despite these histochemical and immunohistochemical staining methods being widely used, SA-β-gal activity is not only present in senescent cells [231,232]. Other indirect biological markers of cell senescence are the expression of genes such as the p16, p21, p53 and _ϒ_H2AX that tend to be increased due to the DNA damage associated with the cellular senescence processes [17,31,233,234,235]. The cell cycle regulators p16, p21, p53 and lamin B1 are generally evaluated using histochemistry, immunohistochemistry and immunoblotting [199,230,231], while the DNA damage marker γH2AX is regularly assessed either by microscopy or flow cytometry upon immunofluorescence staining [227,231]. Heterochromatin markers, namely the HP1 and H3K9me, are other molecular markers of SAHF, which also suggest the occurrence of cellular senescence [235]. SAHF formation is preceded by senescence-associated decondensation satellites (SADs), which are also extensively related to cell senescence [236], and can be analyzed by microscopy following immunofluorescence [237]. Adding to this, the expression and secretion of senescence-associated inflammatory and proteolytic factors (SASP) also allow senescent cell detection [238], using the enzyme-linked immunosorbent assay (ELISA) [231]. Importantly, the majority of senescent cells secrete IL-1α, IL-1β, IL-6, IL-8, IL-18 and TNF-α as SASP elements [21,238,239], and these have already been clinically validated as risk factors for the development of CVD [240,241]. On the other hand, ROS-sensing dyes have been reported as being effective for sorting senescent cells [242]. Moreover, the secretion of inflammatory cytokines by senescent cells is supported by the recycling provided by an up-regulation of autophagy [243]. As such, the autophagosome marker microtubule-associated protein light chain 3 (LC3) and the autophagic substrate p62 analysis by immunoblotting is generally used to assess autophagy as a senescence marker [244]. In addition, leukocyte absolute telomere length is also considered a senescence biomarker and can be determined by PCR, FISH or even Southern blot [245,246]. Concerning senescent phenotypes, cellular granularity should also be considered. Cellular granularity corresponds to dense particles that are frequently accumulated in the cytoplasm of senescent cells. Although the senescent granule composition varies depending on the cell type and is difficult to determine in research or clinical approaches, they are known to be composed mainly of lysosome-containing lipofuscins [247]. Beta-amyloid granules [248], secretory vesicles [65] and glycogen granules [249,250,251] are other elements that also contribute to increasing the granule content of senescent cells. While the exact composition of senescent granules requires fine and time-consuming characterization, the quantitative analysis of their accumulation can be easily performed using the side scatter parameter (SSC) during flow cytometry and can therefore be regularly used as an additional senescent cell marker [249]. Remarkably, the lysosomal senescence-associated glycoprotein (SAGP) was recently recognized to be a biomarker of cellular senescence in CVD, but the associated molecular mechanisms still need to be elucidated. It is important to stress that cellular senescence can be induced by several stressing agents in a cell-type dependent manner, justifying the variability of the senescent phenotypes, factors that strongly contribute to the lack of a universal gold-standard senescence marker [46]. Therefore, different parameters should be used to ensure that a cell is senescent, since a single parameter is not sufficient to make this kind of statement. Table 3 shows a comparison of the typical pattern of exhibition of a number of markers by different types of cells, including vascular and post-mitotic senescent cells, reinforcing the need to identify specific markers of senescent cells.

## 6. Senescence-Mediated Therapeutic Strategies for Cardiovascular Disorders

It is well known that chronic diseases, including atherosclerosis, diabetes, osteoarthritis and dementia, are intrinsically related with aging [252]. In addition, these age-associated diseases are known to be associated with pathological cellular processes including chronic inflammation, organelle dysfunction and cellular senescence [11]. Indeed, senescent cell accumulation in tissues is recognized to compromise the specific physiological functions of the affected organ [253]. For instance, senescent cell accumulation in the aorta has been shown to be associated with vascular hyporeactivity, poor vasomotor function and atherosclerosis [74]. Importantly, cell senescence is able to drive arteriosclerosis independently of aging in human progeria syndromes, characterized by marked early senescent cell accumulation, increased atheroma incidence and a consequent elevated risk for CVD [254,255,256]. Notably, strategies to promote the clearance of senescent cells from the arterial vessels have been shown to improve the typical age-related vascular phenotypes and therefore may be a potential therapeutic intervention to reduce morbidity and mortality from CVD [74]. From a clinical perspective, senescent cells exist in moderate amounts in aged tissues and are core players in various pathological processes, thus being an attractive target to prevent and treat age and senescent cell-associated CVD. Indeed, efforts have been made to develop senolytic drugs, which selectively remove senescent cells by apoptosis [257]. The tyrosine kinase inhibitor dasatinib and the flavonoid quercetin were the first senolytic drugs to be reported [257]. Initial evidence suggests that senolytic therapies could be administered intermittently and that therapy resistance is unlikely to occur. Remarkably, senescent cells secrete pro-apoptotic factors that neutralize nearby cells, but to which they are resistant, due to the activation of anti-apoptotic pathways. In line with this, it was found that both dasatinib and quercetin, amongst others, are capable of regulating the anti-apoptotic pathways of senescent cells facilitating their elimination in rodent and human cell cultures, fresh human tissue explants and mouse models of aging [14,253,257,258]. However, the potential of senolytic drugs to regulate anti-apoptotic pathways varies depending on the origin of the senescent cells, and as such the newly discovered drugs do not show the same level of efficiency against all senescent cell types. Promising therapeutic approaches are therefore likely to arise from the targeting of multiple anti-apoptotic signaling pathways, with complex drug cocktails that may be able to neutralize a wider array of senescent cell types. Similar to that occurring in current cancer treatment, these multidimensional treatment modalities are also likely to require exposition to lower levels of each specific drug and contribute to attain a therapeutic synergism.

In addition, even though scarce adverse effects have been perceived in mice exposed to senolytics, adverse effects of some drugs have been documented [253]. The drug navitoclax (ABT263), an anti-cancer agent with senolytic activity, regularly causes neutropenia and thrombocytopenia [253]. Senolytic treatment also poses other hidden challenges; while it can neutralize well-established senescent cells carrying potentially oncogenic mutations and thus hinder cancer development, senolytic approaches have also been reported to affect the protective and anti-carcinogenic cellular senescence mechanisms [258]. Nevertheless, mice treated with a combination of dasatinib and quercetin exhibited increased survival and health span. The administration of senolytics in mice also relieved the onset of physical disabilities, delayed chronic disorders associated with aging and reduced survival upon senescent cell transplantation [258]. In addition, exposure to dasatinib and quercetin was shown to boost vasomotor function and to diminish aortic calcification in aged and hypercholesterolemic mice, respectively [74], significantly improving cardiac function in aged mice [257]. Stimulation of cardiac progenitor cells in aged hearts and increased cardiomyocyte proliferative capacity were also reported upon senescent cell elimination in aged mice, both using pharmacological approaches or in genetic senolytic models [38]. Pharmacological and genetic senolytic models supported a link between senescent cell depletion, inhibition of heart fibrosis and increased cardiomyocyte proliferative expression profile [37]. Furthermore, senescent cardiomyocyte elimination via administration of ABT263 improved myocardiac remodeling and the overall survival rate in a myocardial infarction mouse model [259]. These findings demonstrate that senolysis is able to reverse phenotypic changes associated with aging, namely through the reversion of age-associated cardiac dysfunction and by promoting its regenerative capacity, which reinforces senolysis as a potential strategy to treat CVD [253,260]. In addition, cardiac glycosides were recently identified as senolytic compounds, with strong potential to be used in effective treatments against age-associated disorders [261,262]. The HSP90 chaperone inhibitors were also identified as a novel class of senolytics [263], amongst which 17-DMAG was found to ameliorate atherosclerosis in mice [264], potentially due to its senolytic activity. In addition, 2-deoxy-D-glucose (2DG), a glucose analog that inhibits ATP synthesis leading to cell cycle arrest and cell death, was shown to have a senolytic effect on senescent vascular smooth muscle cells. The senolytic effect of 2DG potentially depends on the increased metabolic activity of senescent cells, which may have an impact on the progression of atherosclerosis [76]. Although some senolytics have been clinically approved or are already in clinical trials for oncologic diseases, idiopathic pulmonary fibrosis and chronic kidney disease [265], studies with senolytic drugs have only been performed in animal models of disease in the field of atherosclerosis, and clinical trials are currently awaited.

On the other hand, distinct studies have suggested that suppressing cellular senescence is another potential strategy to develop therapies for cardiovascular disorders. Amongst these, activation of Sirtuin1 (SIRT1) signaling has been consistently reported as being capable of inducing suppression of cellular senescence. Specifically, SIRT1 activation mediated by the polyphenol resveratrol was shown to prevent both arterial wall inflammation and stiffening in nonhuman primates [266]. Likewise, SIRT1 specific activator SRT1720 was shown to alleviate hypertension and arterial stiffness in mice [267]. Important agreeing studies revealed down-regulated SIRT1 expression in VSMCs of patients suffering from abdominal aortic aneurysm, while SIRT1 activation was accompanied by the inhibition of cell senescence and diminished vascular inflammation [268]. Accordingly, calorie restriction was found to be associated with SIRT1 activation in VSMCs and decreased prevalence of abdominal aortic aneurysm [269]. In agreement, other related studies have also demonstrated that the suppression of VSMC senescence is mediated by SIRT1 signaling pathways [270,271]. Other clinical treatments for CVD include the drug pioglitazone that stimulates telomerase activation, attenuating EC senescence [272].

Alternative approaches to target senescent cells are also being investigated. These include vaccines and other modulators of the immune system to hasten the elimination of senescent cells, as well as toxin delivery through senescent cell-recognizing technologies. The potential of drugs such as metformin and rapamycin (sirolimus) to inhibit the SASP, diminish the pro-inflammatory drive and the damage caused by activated senescent cells is also being explored [253]. Specifically, metformin [273] and rapamycin [173,274,275] were shown to alleviate CVD. These drugs are considered senostatic or senomorphic drugs, as they inhibit the SASP, preventing the progression of cell senescence, without inducing the death of senescent cells [276]. Regarding the energy-sensing mTOR pathway, its inhibition by rapamycin led to extended mice lifespan, alleviated senescence and exhibited anti-atherosclerotic effects [178,179]. Moreover, decreased arterial expression of the senescence marker p19, age-associated vascular dysfunction and oxidative stress reversal were observed upon dietary intake of rapamycin [277]. A recent study also revealed that mTORC1 inhibition mediated by rapamycin strongly weakened replicative senescence in human cardiac progenitor cells [177], which may open avenues to develop novel therapies. Moreover, statins were reported to inhibit the SASP and participate in the regulation of the cell cycle and telomerase, delaying both EC and T-cell senescence [278]. Interestingly, modulation or removal of senescent cells targeting the marker SAGP was recently postulated as a potential therapy for atherosclerosis.

Importantly, advances have been provided in the development of natural-based bioactive compounds with potential anti-senescence properties, termed nutraceuticals [279]. Polyphenols, abundantly present in plants, exhibit antioxidant and anti-inflammatory properties, thus being potential senostatics by counteracting pro-oxidant and pro-inflammatory signaling in senescent cells [280]. Interestingly, resveratrol, already shown to exhibit cell senescence suppressor properties in the context of cardiovascular complications [266], is found amongst these compounds [280]. As the development of nutraceutical-based senolytics and senostatics to alleviate age-related diseases is being pursued [279,280], hope is put into these additional potential strategies to face CVDs.

On the other hand, mesenchymal stem cell (MSC)-based therapies are also being developed to face senescence-associated CVDs. MSCs are multipotent cells with multi-differentiation potential and low immunogenicity [281]. Notably, clinical trials on MSC transplantation are currently ongoing, suggesting cardiac improvements in subjects with heart failure secondary to ischemic cardiomyopathy [282].

Considering the overall gathered evidence of the potential treatment regiments in the mitigation of pathological cell senescence overruling atherosclerosis (Table 4), great expectation is placed on clinical trials with senolytic agents that are currently being waited for. Clinical trials assessing the safety and efficacy of senolytic drugs may allow further progress to the exciting stages of patient drug administration and provide novel therapies to prevent the development of cell senescence and CVD [20].

## 7. Conclusions

Cell senescence is a complex process involving various cellular components and signaling pathways in a response to diverse physiological events. Cellular senescence is also intrinsically involved in the pathological process of atherosclerosis, regulating various morphological and functional alterations with dramatic consequences for atheroma progression due to SASP.

Despite the great therapeutic advances made to prevent and treat atherosclerosis and CVD, the acute manifestations of these pathologies continue to be associated with an enormous socio-economic burden and the main cause of death worldwide. Targeting senescent cells via a variety of therapeutic approaches, including senolytics and senostatics, has emerged as a promising strategy to mitigate atherogenesis. In this context, targeting the subjacent marked dysfunctional organelles that arise during cellular senescence, namely senescent lysosomes and mitochondria, is key to counteracting senescence-mediated atherosclerotic events. Therefore, better clarification of the role of senescence, the molecular mechanisms involved, and the specific implications of senescent organelles in atherosclerosis is urgently needed, as the challenging clinical trials, so that the best therapeutic approaches can be outlined in the near future.

## Figures and Tables

**Figure 1 cells-09-02146-f001:**
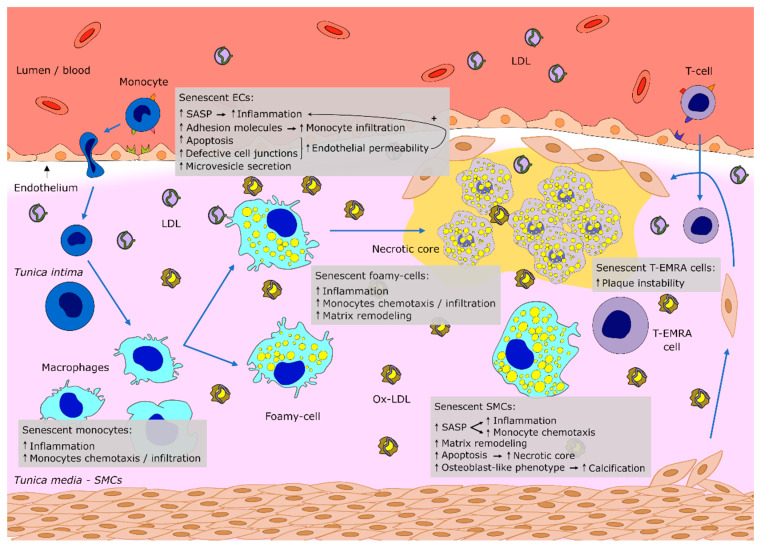
Schematic representation of an artery wall section showing the main events driven by senescent vascular cells, which contribute to the atheroma progression and consequently vascular disease development. The typical senescence-associated pro-inflammatory phenotype (SASP) fuels inflammation, monocyte chemotaxis and endothelial infiltration, eased by increased endothelial permeability. The accumulation of oxidized lipids in the vascular wall and consequent foam cell formation occur along with the recruitment of vascular SMCs from the *media* to form a fibrous cap, which progressively becomes destabilized. Senescent immune cells found at lesion sites contribute to plaque instability, acting synergistically with calcification events to increase the vessel vulnerability. ECs—endothelial cells, SMCs—smooth muscle cells, LDL—low density lipoprotein, SASP—senescent-associated secretory phenotype.

**Figure 2 cells-09-02146-f002:**
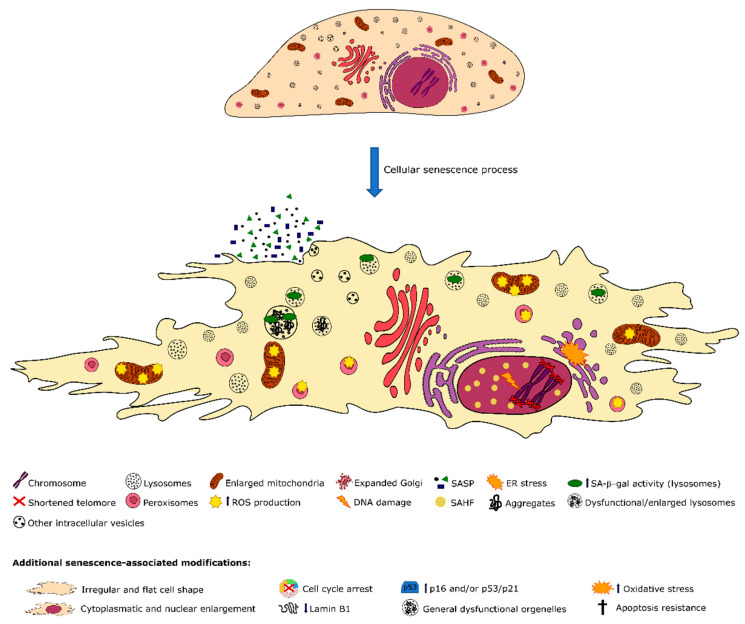
Scheme illustrating the main morphological and related functional differences between a normal proliferating and a senescent cell. Emphasis is given on the multiple organelle modifications occurring during the cellular senescence process, including enlarged lysosomes enclosing lipid and protein aggregates associated with the decline in degradative capacity. Dysfunctional mitochondria and peroxisomes lead to increased ROS generation. An expanded ER and Golgi apparatus are also observed, along with nuclear changes, including severe chromatin condensation. ER—endoplasmic reticulum, SASP—senescent-associated secretory phenotype, ROS—reactive oxygen species, SAHF—senescent-associated heterochromatin foci, SA-β-gal—senescent-associated β-galactosidase.

**Table 1 cells-09-02146-t001:** Major SASP components in different senescent cell types.

Cell Type	SASP Components
Human fibroblasts	Interleukins: IL-1b, IL-6, IL-7, IL-8, IL-11, IL-13, IL-15
	Chemokines and cytokines: GRO-a, GRO-b, GRO-g, MCP-1, MCP-2, MCP-4, MIP-1a, MIP-3a, HCC-4, Eotaxin-3, MIF, GM-CSF, OSM, Leptin
	Growth factors and regulators: bFGF, FGF-7, SCF, PIGF, HGF, IGFBP-1, IGFBP-2, IGFBP-4, IGFBP-6, Angiogenin
	Proteases and regulators: TIMP-2
	Receptors and ligands: ICAM-1, ICAM-3, sTNF RI, sTNF RII, TRAIL-R3, uPAR, Axl, OPG, Fas/TNFRSF6, sgp130
Human epithelial cells	Interleukins: IL-1a, IL-1b, IL-2R-a, IL-6, IL-8
	Chemokines and cytokines: GRO-a, GRO-b, GRO-g, MCP-1, MCP-4, MIP-1a, MIP-3a, HCC-4, CTACK, I-TAC, GM-CSF, GCP-2, ENA-78, MIF
	Growth factors and regulators: bFGF, PIGF, VEGF, PDGF-BB, IGFBP-2, IGFBP-6, Amphiregulin, Thrombopoietin, Angiogenin
	Proteases and regulators: TIMP-1, TIMP-2
	Receptors and ligands: ICAM-1, IL-6R, uPAR, EGF-R, sTNF RI, sTNF RII, GITR, OPG, sgp130
Human epithelial cancer cell lines	Interleukins: IL-1b, IL-6, IL-8
	Chemokines and cytokines: GRO-a, GM-CSF
	Growth factors and regulators: IGFBP-2

SASP—senescence-associated secretory phenotype.

**Table 2 cells-09-02146-t002:** Methodologies used for detection of the most common senescent cell biological markers.

Biomarkers	Methods Used for Detection
Proliferation markers	Flow cytometry
Cell morphology	Phase-contrast microscopy
SA-β-Gal	Histochemistry, immunohistochemistry
p16, p21, p53, lamin B1	Histochemistry, immunohistochemistry, immunoblotting
_ϒ_H2AX	Immunohistochemistry, flow cytometry
SADs, SAHF	Immunohistochemistry
SASP	ELISA
ROS	Flow cytometry, microplate readers, microscopy
Autophagic markers	Immunoblotting
Leukocyte absolute telomere length	PCR, FISH, southern blot
Cellular granularity	Flow cytometry

SA-β-Gal—senescence-associated-β-galactosidase; SADs—senescence-associated decondensation satellites; SAHF—senescence-associated heterochromatin foci; SASP—senescence-associated secretory phenotype; ELISA—enzyme-linked immunosorbent assay; ROS—reactive oxygen species; PCR—polymerase chain reaction; FISH—fluorescence in situ hybridization.

**Table 3 cells-09-02146-t003:** Comparison of different markers exhibited by senescent vascular and post-mitotic cells as well as quiescent cells.

Cellular Markers	Senescent Vascular SMCs	Senescent Vascular ECs	Senescent Monocytes	Post-Mitotic Cells	Quiescent Cells
Cell morphology alterations	+	+	+	+	-
Cell cycle arrest	+	+	+	+	reversible
Oxidative stress	+	+	+	+	-
Apoptosis resistance	+	+	+		+
SASP	+	+	+	non-canonic	-
DNA damage	+	+	+	+	+

SMCs—smooth muscle cells; ECs—endothelial cells; SASP—senescence-associated secretory phenotype.

**Table 4 cells-09-02146-t004:** Senescence-based therapeutic approaches for atherosclerosis and cardiovascular disorders.

Therapeutic Strategies	Drugs	Model Species/Clinical Trials	Mechanism of Action
Senolytics	Dasatinib, quercetin, navitoclax	Mouse models	Induction of cellular apoptosis
	Cardiac glycosides	Human and mouse models	
	2DG	Human models	
Cellular senescence suppressors	Resveratrol	Non-human primate models	Inhibition of cellular senescence
	SRT1720	Mouse models	
	Pioglitazone	Human, mouse and bovine models	
Senostatics	Rapamycin, metformin, statins	Human and mouse models	SASP inhibition
MSC-based	–	Early phase trials	Cell engraftment

2DG—2-deoxy-d-glucose; SASP—senescence-associated secretory phenotype; MSC—mesenchymal stem cell.

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
