# Peer review of "Cell Senescence, Multiple Organelle Dysfunction and Atherosclerosis"

_cells, 2020, doi:10.3390/cells9102146_

Round 1

Reviewer 1 Report

The manuscript entitled ‘Cell senescence, multiple organelle dysfunction and atherosclerosis’ discusses the senescence-related alterations in major intracellular organelles and analyzes the role of different cell types in senescence and atherogenesis. The review provides a thorough survey of the relevant studies in a well-organized way. However, there are a few suggestions that may improve the presentation of the readership of the manuscript.

  • Since the review focused on the endothelial senescence, a detailed mechanistic insight of the endothelial senescence should be provided.
  • In the heart, cardiomyocytes contribute maximum to the cardiac mass as well as the senescent cell population. Cardiomyocytes also contribute maximum to the SASP and thus associated pathophysiology. Therefore, a detail about the cardiac senescence and it’s pathophysiological should be provided, which is almost missing in the review.
  • Also, include the studies, which target the senescence-associated genes to improve the cardiac protection and cardiac function. Like:

(i) Inhibition of Senescence-Associated Genes Rb1 and Meis2 in Adult Cardiomyocytes Results in Cell Cycle Reentry and Cardiac Repair Post-Myocardial Infarction. Alam et al, 2019.

(ii) Meis1 regulates the metabolic phenotype and oxidant defense of hematopoietic stem cells. Kocabas et al. 2012.

  • The introduction section is missing the relevant references at multiple points.

Reviewer 2 Report

I thank the authors for their very interesting work which is elucidating the topic of cell senescence and atherosclerosis. The review is very thorough.

Minor comments:

1) I would add the methods. At present, it is not clear to me, how this review was done. Which articles, from which time periods were included? How and where was the search for relevant articles performed?

2) I would like to stress, that atherosclerosis is not responsible for all strokes occuring. In some patients with strokes, atherosclerosis is not present or only present just by chance. I would change the working, if strokes are mentioned.
